# Real-time dynamics and structures of supported subnanometer catalysts via multiscale simulations

Yifan Wang[1,2,5], Jake Kalscheur[1,2,5], Ya-Qiong Su[3,4,5], Emiel J. M. Hensen [ID] [4✉] & Dionisios G. Vlachos [ID] [1,2✉]

Understanding the performance of subnanometer catalysts and how catalyst treatment and exposure to spectroscopic probe molecules change the structure requires accurate structure determination under working conditions. Experiments lack simultaneous temporal and spatial resolution and could alter the structure, and similar challenges hinder first-principles calculations from answering these questions. Here, we introduce a multiscale modeling framework to follow the evolution of subnanometer clusters at experimentally relevant time scales. We demonstrate its feasibility on Pd adsorbed on $CeO_2(111)$ at various catalyst loadings, temperatures, and exposures to CO. We show that sintering occurs in seconds even at room temperature and is mainly driven by free energy reduction. It leads to a kinetically (far from equilibrium) frozen ensemble of quasi-two-dimensional structures that CO chemisorption and infrared experiments probe. CO adsorption makes structures flatter and smaller. High temperatures drive very rapid sintering toward larger, stable/metastable equilibrium structures, where CO induces secondary structure changes only.

[1] Department of Chemical and Biomolecular Engineering, 150 Academy St., University of Delaware, Newark, Delaware, DE 19716, United States. [2] Catalysis Center for Energy Innovation (CCEI), RAPID Manufacturing Institute, and Delaware Energy Institute (DEI), 221 Academy St., University of Delaware, Newark, Delaware, DE 19716, United States. [3] School of Chemistry, Xi'an Key Laboratory of Sustainable Energy Materials Chemistry, State Key Laboratory of Electrical Insulation and Power Equipment, Xi'an Jiaotong University, Xi'an 710049, China. [4] Laboratory of Inorganic Materials and Catalysis, Department of Chemical Engineering and Chemistry, Eindhoven University of Technology, P.O. Box 513, 5600 MB Eindhoven, The Netherlands. [5] These authors contributed equally: Yifan Wang, Jake Kalscheur, Ya-Qiong Su. ✉email: e.j.m.hensen@tue.nl; vlachos@udel.edu

Supported platinum group metal (PGM) catalysts are used in the catalytic converter of vehicles to convert exhaust gases. The development of low-cost, high-performing catalysts will continue for achieving emission standards[1]. Supported single-atom catalysts (SACs) and subnanometer clusters offer unique catalytic properties for several industrial relevant chemistries[2–5], and their high dispersion (size $n$, $n \sim \leq 30$) exposes nearly all metal atoms for catalysis. Great efforts have been devoted to developing thermally stable, atomically dispersed SACs, and subnanometer cluster catalysts[6]. Loss of dispersion and catalyst deactivation could occur over time due to sintering via (1) individual particle migration and coalescence and (2) Ostwald ripening, where single atoms detach from a particle, diffuse on the support, and attach to another particle[7,8]. Sintering depends strongly on metal loading and environmental factors, such as temperature, atmosphere, and metal-support interactions[9]. To maintain dispersion, metal loadings are typically very low, and this makes in-situ and operando characterization very challenging. Aside from the detection limit, the time resolution of experimental methods is medium to long and this precludes insights into fast catalyst dynamics[1,10,11]. Multiple experimental techniques are used to characterize the catalysts but their effect on the catalyst structure and dynamics, e.g., upon exposure the metal to CO for infrared (IR) measurements, is unknown. The catalyst activity and selectivity are dictated in part from their structure controlled by metal-metal, metal-adsorbate, and metal-support interactions[11,12]. Mastering the structure evolution at the atomic scale under working conditions is key to understanding structure-reactivity relations and discovering new catalysts.

Multiscale simulations can complement and potentially extend the spatial and temporal resolution of experiments. However, this is a daunting task. The typical horsepower of computational catalysis, density functional theory (DFT), captures interactions, and energies but is limited to tiny scales at 0 K and performs only local optimization during relaxation. Consequently, DFT is incapable of predicting complex structures and dynamics. Optimization algorithms can systematically find global and local minima in the energy landscape but need an accurate parameterization of the force field. During optimization, the force field can be computed by "calling" the DFT "calculator," as done in ab initio molecular dynamics (AIMD). However, DFT-based optimization is impractical due to the high computational cost and the inability to perform DFT calculations for large clusters. It is then not surprising that computations optimize a single, small particle size[13–17]. Integrating efficiently transferable (in terms of size) force fields, computed via DFT, with optimization methods is the first crucial step toward predicting equilibrium structures. Predicting dynamics, relevant time scales, distribution of sizes and shapes, and whether the system reaches the equilibrium state is vital for understanding catalysts and is a harder problem. MD captures the dynamics but is still limited to nanosecond time scales[18,19]. There is a clear need for a general, transferable, and interpretable computational framework to predict structures operando (under working conditions).

Here, we study the structure evolution under experimentally relevant conditions using multiscale kinetic Monte Carlo (KMC) simulations. Specifically, we investigate the effect of temperature, CO partial pressure, metal loading, and initial catalyst state on catalyst dynamics under experimentally relevant time scales. We choose Pd ($n = 1$–$40$) on $CeO_2$(111) in a CO atmosphere as a case study, a catalyst commonly used in automobile catalytic converters. CO is a reactant (product) in multiple chemistries and the most common probe molecule in infrared (IR) spectroscopy[20,21]. Our work focuses on three-dimensional (3D) representations of the clusters instead of conventional 2D projections[22]. The KMC simulation engine incorporates accurate,

efficient, and transferable machine learning (ML)-based Hamiltonians (trained on DFT data) to describe the energetics of cluster isomers and CO adsorption at all sizes and sites. The kinetics of Pd single atom and small clusters on ceria, consistent with the thermodynamic data, are also included. The results reveal that CO adsorption flattens the structures to bilayer and shrinks them somewhat; long-time cluster growth from dispersed single atoms or smaller clusters gets kinetically frozen at far-from-equilibrium states at low temperatures. We demonstrate that thermodynamic stability calculations are insufficient to study catalyst structure. The developed computational framework depicts a dynamic picture of subnanometer catalysts, potentially bridging the gap between the first-principles calculations and experiments.

## Results and discussion

The multiscale framework (Fig. 1a–c) allows modeling of metal clusters that are bare or exposed to an adsorbate used for characterization, e.g., CO in IR, or an intermediate created by the surface chemistry and does so at an unprecedented computational efficiency. The framework integrates a comprehensive toolset including DFT calculations, genetic algorithm-based structure optimization, ML, and KMC. Details of all sub-models can be found in Methods and prior works[8,23]. We first investigate the electronic and structural characteristics of $Pd_n$ clusters on $CeO_2$(111) and the CO adsorption energies on various sites using DFT. DFT calculations supply metal cluster structures and energies. To overcome the computational cost of DFT, we train ML-based Hamiltonian energy models (Fig. 1a) on this DFT data as they represent an efficient structure-to-energy mapping. We develop two ML models using cluster expansion[24,25] for pristine Pd clusters and the CO adlayer and represent the 3D structures (Fig. 1d) on lattices. The lattices in Fig. 1e, f consist of well-defined sites for metal atoms (Pd) and adsorbates (CO), respectively. Combined with the Hamiltonian, Monte–Carlo-based structure optimization algorithms (Fig. 1b), such a cluster genetic algorithm (CGA), predict the low energy structures at finite temperature. Active learning improves the model accuracy by passing the low energy structures for new DFT calculations and retraining the model iteratively. Next, we enumerate the possible elementary events, including diffusion of Pd single atoms and small clusters and CO adsorption/desorption, and compute their barriers in DFT. The Hamiltonian models and barriers parameterize the on-lattice KMC (Fig. 1c) and enable thousands of on-the-fly calculations. Finally, the KMC engine executes key events and computes their transition rates to simulate the evolution of structures as a function of time.

We explore four experimental conditions: CO partial pressure $P_{CO}$, temperature, metal loading, and initial catalyst state. We perform simulations at typical SAC CO oxidation experiments[2,26,27] and IR measurements, namely $P_{CO} = 0$ (the bare Pd case), 0.1 bar, and 1 mbar, at temperatures of 300 and 700 K. The initial catalyst state and the degree of dispersion (SACs vs. subnanometer clusters vs. nanoparticles or combinations) can change under pretreatment, characterization, and reaction. We run simulations starting from dispersed single atoms and ensembles of 3D $Pd_n$ ($n = 4$ and higher) to explore the effect. The Pd coverage on the base layer ranges from ~0.009 to 0.1, corresponding to a metal loading of 0.2 to 1.1 wt% (we convert the metal loadings to lattice coverages in Supplementary Note 8), spanning from high dispersion[6,28,29] to more typical noble catalyst loadings. All KMC simulations, unless stated otherwise, are run on a 20 by 20 lattice with periodic boundary conditions. Each simulation is repeated three times with different random seeds. The simulation time clock is advanced to at least 60 s for 300 K (1 s for 700 K) and often much longer, comparable to

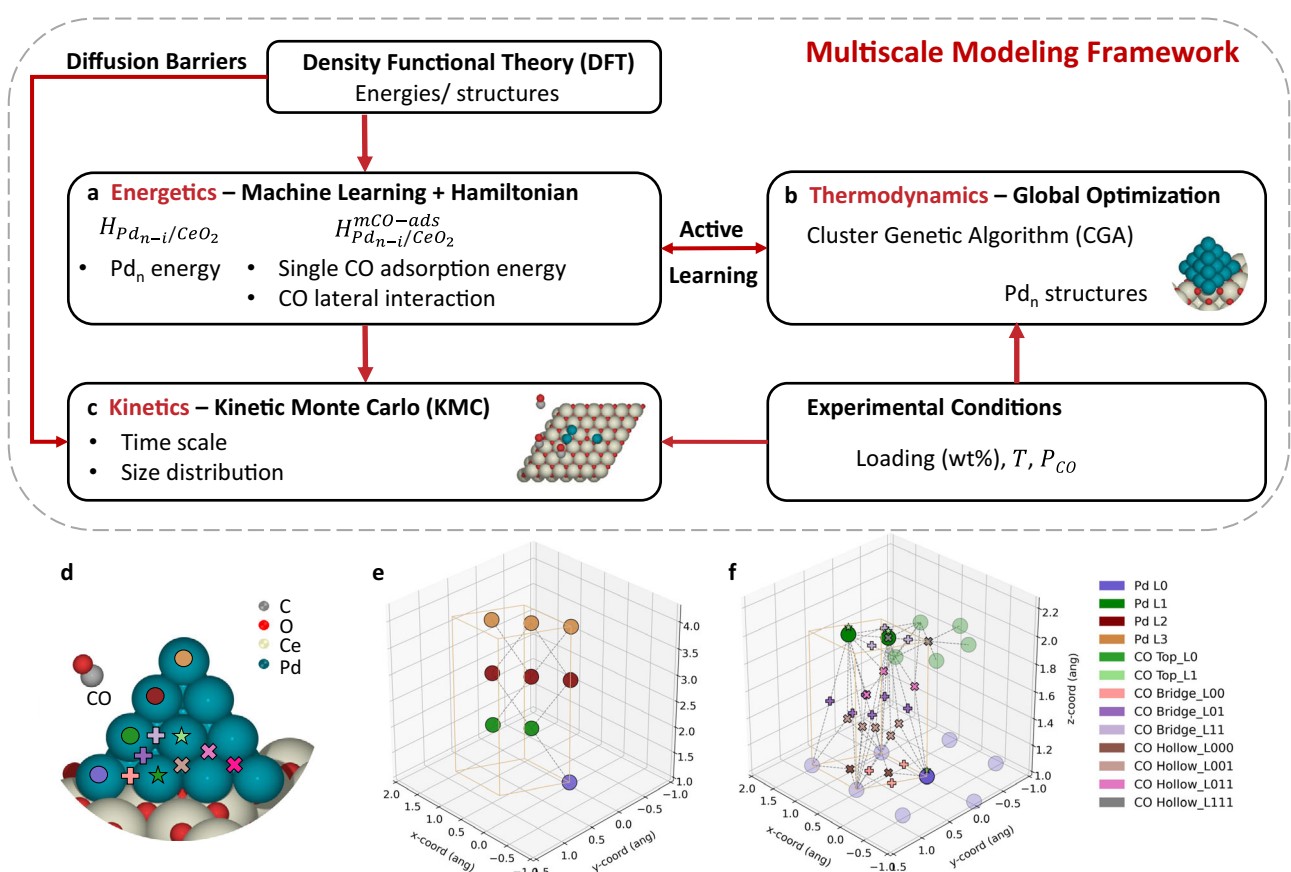

**Fig. 1 Multiscale modeling framework for subnanometer catalysts. a** Machine-learning-based Hamiltonians via cluster expansion for the bare metal (Pd) clusters and adsorbate (CO). The scheme is iterated until convergence using active learning. **b** Monte-Carlo-based structure optimizations: bare clusters (CGA-Metropolis). **c** Kinetic Monte Carlo (KMC) simulations for cluster dynamics. Color code: gray, C; red: O; light yellow, Ce; cyan, Pd. **d** Structure visualization of $Pd_{20}$ on $CeO_2(111)$. Labeled Pd sites and CO adsorption sites on a facet. Color code: gray, C; red, O; light yellow, Ce; cyan, Pd. **e** Lattice of the bare $Pd_n$. Pd sites are labeled according to their distance from the support (Pd L0, L1, L2, or L3; Ln refers to the layer number with respect to its distance from the support, i.e., the base, second, third, fourth layer and so on). **f** Lattice of the $Pd_n$-CO. CO adsorption sites indicate the adsorption type (top, bridge, or hollow) and the layer number of the neighboring Pd atoms, e.g., CO Top_L0 (CO on top of Pd on the layer on the support), Bridge_L01 (CO on bridge site with one Pd on L0 and one Pd on L1), Hollow_L011, and so on. The unit cell boundary is in orange and the nearest neighbor connections in dotted lines. The two-lattice representation allows very efficient computations.

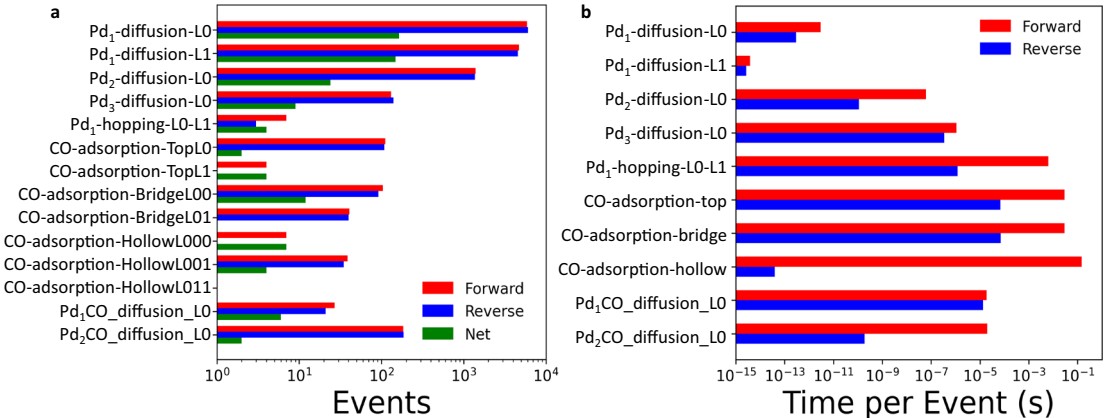

**Fig. 2 Illustration of disparity in time scales of microscopic events. a** Scaled event frequency and **b** actual average time per event (without prefactor scaling) in a KMC simulation starting from a 0.05 coverage of single atoms at 300 K and $P_{CO} = 0.1$ bar at long times (over 168 h).

experimental time scales. For each complex represented as $Pd_nCO_m$, a Gaussian process (GP)[30] model is trained from triplicate runs. The GP models, using a standard Matérn kernel, are well suited to time-series data analysis and reflect the averages and uncertainties as a function of time[31].

The rates of elementary events reveal the critical steps and are illuminating (Fig. 2). Prefactor rescaling is a common multiscale technique in KMC to advance the time clock to long times[32–35] without impacting the results (see Supplementary Note 7). After prefactor rescaling, small cluster ($n \leq 3$) diffusion is at least 100

times faster than other events (Fig. 2a). Small CO-Pd cluster diffusion is also rapid but slower than bare cluster diffusion ($n \leq 3$), i.e., CO adsorption slows down the Pd diffusion and the overall dynamics of small clusters (the barriers are shown in Supplementary Tables 2 and 3). However, for larger two-layer clusters, such as $Pd_4\_3d$, the diffusion barrier for the bare cluster (1.35 eV) is higher compared to the one when CO is attached to the cluster (1.03 eV), i.e., adsorbed CO makes their diffusion faster. Single Pd atoms hop from the base layer to one above slowly, i.e., the 2D to 3D restructuring transition is also slower. CO adsorption occurs on various sites with varying frequencies. Adsorption events on single atoms (Top_L0), dimers (Bridge_L00), and in between layers L0 and L1 on larger clusters (Bridge_L01 and Hollow_L001) are fast and quasi-equilibrated (a partial equilibrium index (PEI)[36] of ~0.5). The unscaled events (the actual average time per event is shown in Fig. 2b) follow the reverse relative order. The time scales are revealing: $Pd_1$ diffusion on L0 and L1 occurs at picoseconds ($10^{-12}$ s). Diffusion of small clusters on the base plane, such as $Pd_2$ and $Pd_3$, occurs between nanoseconds ($10^{-9}$ s) and microseconds ($10^{-6}$ s). CO adsorption and $Pd_1$ hopping from layer 0 to layer 1 are the slowest, with a time scale between milliseconds ($10^{-3}$ s) and seconds. The latter must overcome a large energy barrier. Our calculations indicate complex single event time scales that depend on size and site and span 14 orders of magnitude.

We show the time evolution of the various species counts (divided by the number of Pd atoms, n) in Fig. 3. The number of CO molecules per Pd atom $\bar{m}_{CO}$, also known as stoichiometry, is a good descriptor for the number of CO molecules adsorbed per Pd atom. We also include the L1/L0 ratio (the number of Pd atoms in the second L1 layer divided by the number in the base L0 layer) as a descriptor of 3D structures and microscopic detail on CO adsorption sites (type of site and binding and layer preference). In Fig. 3a–e, we observe that both CO adsorption and Pd hopping in the first 10 s as $\bar{m}_{CO}$ and L1/L0 ratio increase with time (see more details in Supplementary Fig. 1). Afterwards, CO adsorption continues to ramp up to $\bar{m}_{CO}$ ~1.5. At ~200 s, L1/L0 stabilizes to

~0.2 (one Pd atom on the second layer per four Pd atoms on the base plane, on average); the large confidence intervals indicate substantial fluctuations in the structures, which is expected given the small number of Pd atoms in these tiny clusters. Interestingly, Fig. 3d, e shows that many COs adsorb initially on bridge and hollow sites formed by Pd atoms on the base plane (BridgeL00 and HollowL000) and then desorb upon forming the 3D structure. Between 200 and 400 s, further CO adsorption occurs on the sites near the cluster periphery in between the two layers (HollowL001 and HollowL011). We observe no significant change in the dynamics and structures after 400 s. Throughout, CO is adsorbed preferentially at the bridge and hollow sites than on top sites.

The results expose fascinating phenomena. Importantly, it is feasible to capture experimentally relevant time scales while maintaining atomic resolution. Sintering happens fast, exhibits multiple time regimes, and the evolution completes itself within seconds—tiny, relatively flat clusters with a single atom on the second layer form. In a typical IR experiment, one would expect to see clusters probed by the IR and thus broader spectra than those of single atoms—a topic elaborated further below. The catalyst dynamics is more rapid than that of CO adsorption/ desorption at room temperature. We elaborate on the effect of CO partial pressure below. The CO chemisorption stoichiometry ($\bar{m}_{CO}$) approaches ~1.5–1.6 rather than ~1 typical for large nanoparticles or two for many single metal atom systems[37,38], and thus, its interpretation from experiments is not trivial without calculations.

We visualize the structure evolution for low, medium, and high loadings starting from single atoms in Fig. 4 (initial configurations in Fig. 4a, g, m). We use the mean cluster size ($\bar{n}$) and its standard deviation ($\sigma_n$), the average CO loading $\bar{m}_{CO}$ per Pd atom, and the L1/L0 ratio, to describe clustering, CO adsorption, and 3D clusters, respectively. Starting with bare Pd ($P_{CO} = 0$ bar) and low loadings (Fig. 4d), we observe a single $Pd_8$ cluster in $t = 60$ s. Increasing the loading (Fig. 4j) leads to a distribution of cluster sizes and shapes rather a single cluster with significant

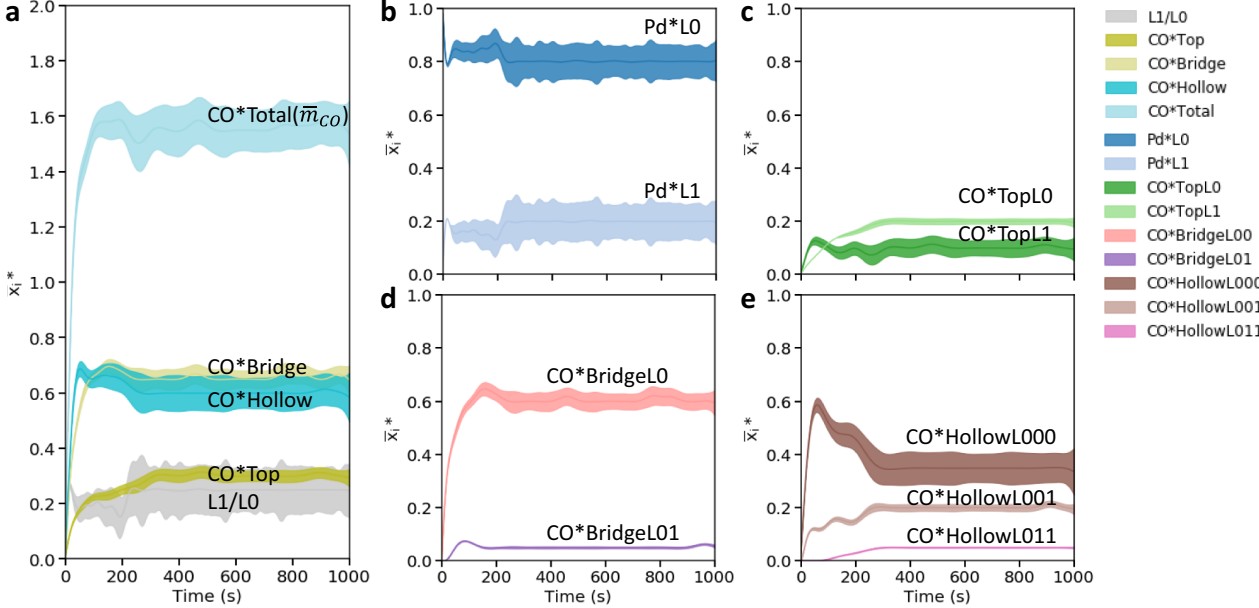

**Fig. 3 Species ratios as a function of time at 300 K exposed to a CO partial pressure $P_{CO} = 0.1$ bar. a** L1/L0 ratio—the number of Pd atoms in the L1 layer to the number in the base L0 layer, $\bar{x}_i^*$—the ratio of total CO adsorbate counts on top, bridge, and hollow sites to the number of Pd atoms vs. time. **b** $\bar{x}_i^*$— the ratio of Pd atom counts on various Pd sites and **c**–**e** CO adsorption adsorbate counts on various CO adsorption sites to the number of Pd atoms vs. time. The means and 95% confidence intervals over the triplicate runs are shown with solid and shaded areas, respectively. A zoomed-in version for the first 60 s is in Supplementary Fig. 1. Simulations starting from a 0.05 coverage of single atoms.

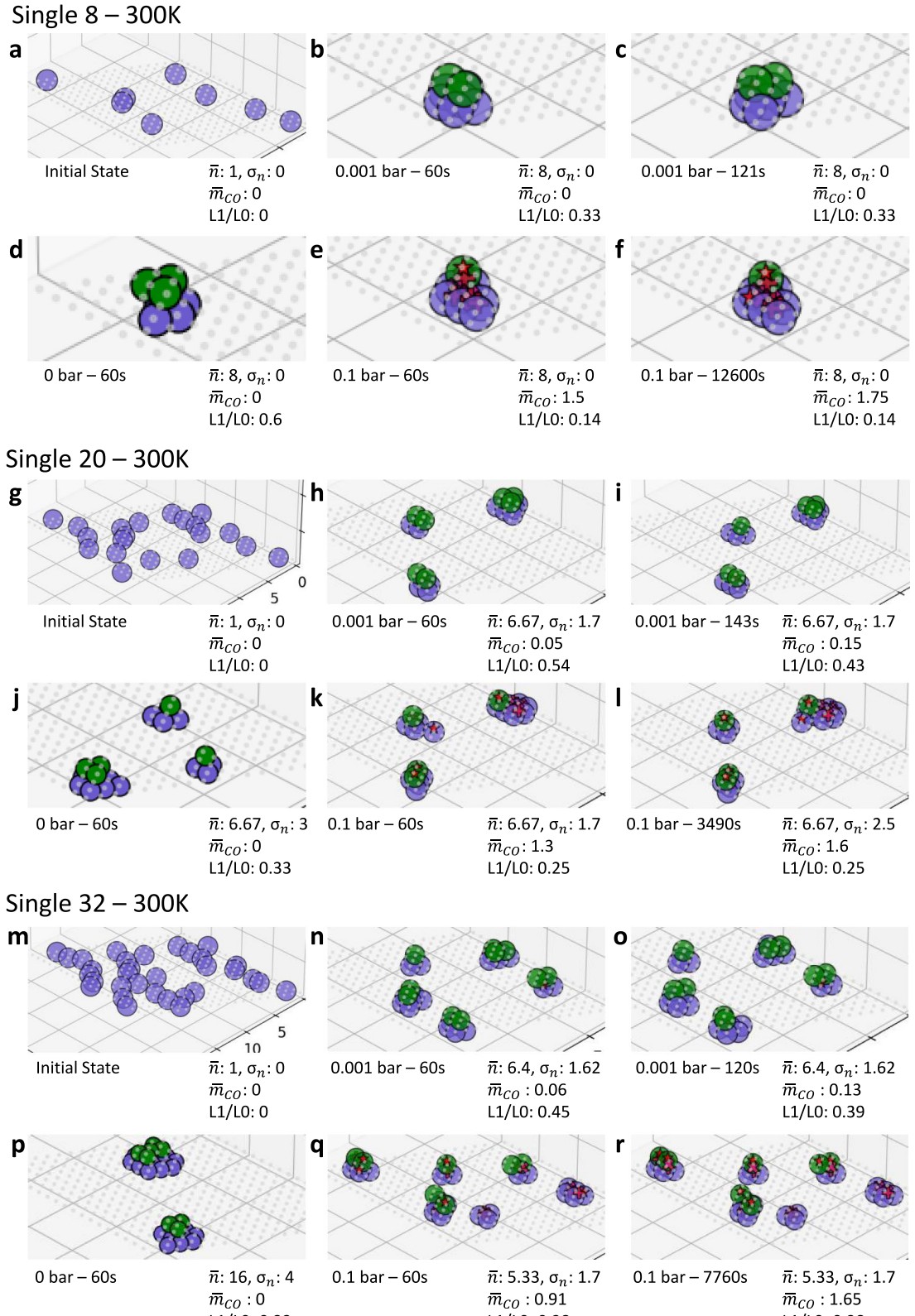

**Fig. 4 Effect of loading and CO partial pressure on catalyst dynamics at 300 K.** An initial state of (**a–f**) 8, (**g–l**) 20, (**m-r**) 32 single atoms (coverages of 0.02, 0.05, and 0.08 or 0.23, 0.57, and 0.92 wt%). For each loading/panel, the initial random configuration is at the top left of each panel with the bare cluster structure underneath. The second column shows snapshots at 60 s and the third for the longest time simulated in the presence of CO, and the top (bottom) row is for low (high) $P_{CO}$. Color code: gray, empty lattice sites; red, CO; purple, Pd in the base plane L0, green: Pd in the second layer L1. Notation: $\bar{n}$ -average cluster size, $\sigma_n$—standard deviation of cluster size, $\bar{m}_{CO}$—the CO loading, L1/L0 ratio—the ratio of the number of Pd atoms in second and base layers.

fluctuations in size and number of Pd atoms in the second layer. Even at high loadings (Fig. 4p), the cluster size is less than 20 atoms (a diameter of 1.2 nm). The bare Pd calculations clearly show that sintering happens without CO. The clusters are always 3D but retain two layers for smaller loadings, i.e., they are quasi-two-dimensional. It is unlikely to sinter single atoms within minutes into large nanoparticles at room temperature.

CO affects the structures. At a low CO pressure (1 mbar) and low loadings, adsorption occurs rarely and slowly after forming 3D structures (Fig. 4c, i); the final structures resemble the bare ones. Counterintuitively, the cluster size is significantly smaller compared to the bare case at higher loadings ($\bar{n} \approx 6$ vs. 16, contrast Fig. 4o from Fig. 4p). At high CO pressure (0.1 bar), CO adsorption is more frequent and $\bar{m}_{CO}$ approaches ~1.5 (Fig. 4f, l, r). The cluster sizes remain small. Compared to the low CO pressure, the L1/L0 ratio decreases to half (0.33 vs. 0.14, 0.54 vs. 0.25, 0.45 vs. 0.28, contrast Fig. 4b, h, n from Fig. 4f, l, r) at each metal loading. Overall, CO renders the clusters smaller, flatter, or even single layer (epitaxial 2D). Dispersed $Pd_1(CO)$ complexes are observed directly. As shown in Fig. 2b, metal atom hopping, and CO adsorption have comparable time scales. In all simulations, we see no significant long-time structure changes after 60 s. Reversible $Pd_1$ diffusion on the base L0 and the second L1 layer keeps occurring (shown in Fig. 2 and Supplementary Movies 1–7) but does not contribute to significant structural changes beyond fluctuations.

Next, we discuss the dynamics after some pre-sintering, e.g., minutes after the synthesis of single atoms, by initializing the calculations with small 3D clusters ($Pd_4$) at 300 K (Fig. 5) at low, medium, and high loadings corresponding to metal loadings 0.23, 0.69, and 1.15 wt%, respectively. The bare clusters remain mostly tetramers, and cluster dynamics are sluggish (Fig. 5j, p). At low CO pressure (1 mbar), a few CO adsorption events occur at long times of 100–1000 s (Fig. 5c, i, o). Increasing the CO pressure to 0.1 bar leads to more CO adsorption (final $\bar{m}_{CO} \sim 1.5$, see time evolution in Supplementary Fig. 2), more structure changes, and heterogeneity in the amount of CO adsorbed (less CO on small clusters). The cluster sizes, even after ~20 min, are smaller than those seen starting from single atoms (contrast Fig. 5f, l, r and Fig. 4f, l, r). This finding is a clear indication that kinetics rather than thermodynamics controls the room temperature structures, i.e., the time scales, the initial synthesis and pretreatment, and the CO partial pressure if significantly high affect the structures in this order. Supplementary Fig. 3 compares the effect of pressure for 20 single atoms and 6 $Pd_4$_3d clusters, and Supplementary Fig. 4 shows the L1/L0 ratio vs. $\bar{m}_{CO}$ of the final configurations from 36 simulations. We observe a negative correlation ($r_{pearson} = -0.77$) between L1/L0 ratios and $\bar{m}_{CO}$ for the single-atom systems, i.e., quasi-2D and 3D clusters accommodate less CO. The single-atom systems are more sensitive in growing vertically to the CO pressure than the $Pd_4$_3d system and a higher CO pressure leads to flatter structures. For the $Pd_4$_3d systems, no strong correlation between L1/L0 ratios and $\bar{m}_{CO}$ is observed ($r_{pearson} = -0.33$). Overall, the results show that small clusters, such as $Pd_4$_3d, are extremely stable once formed (kinetically frozen) at room temperature regardless of the CO pressure.

At higher temperatures, e.g., 700 K, single clusters form for most loadings within a second irrespective of initial starting configuration (results in Supplementary Fig. 5–9), suggesting rapid sintering. $Pd_1$ hops as fast as $Pd_1$-$Pd_4$ diffusion. The clusters experience growth in both the lateral and vertical directions. The long-time structures possess more Pd atoms in the second layer (an L1/L0 ratio > 0.5). Clusters resemble those of independently performed equilibrium structure optimization, for example, $Pd_{20}$-rod (Supplementary Fig. 10d). CO adsorption is discouraged and rarely occurs in the simulation time. KMC simulations are

preoccupied with diffusion, advancing the time clock much slower. Additional CO adsorption could still happen in a longer time but would hardly contribute to the short-time (seconds to minutes) structure change. Small cluster diffusion becomes more frequent, suggesting that particle migration in addition to atom diffusion also plays a role at high temperatures.

The KMC simulations underscore a fascinating dynamic picture for the supported metal subnanometer catalysts. Small clusters form in microseconds, then vertical growth occurs, and finally (or simultaneously at room temperature), CO adsorbs in milliseconds and seconds. Contrary to the traditional reductionist belief, single atoms do not disappear after agglomeration but always co-exist with clusters. Single atoms detach from clusters, diffuse readily on the support, and drive long-time structure changes. At room temperature, since agglomeration is driven by single atom and small cluster diffusion, once the cluster grows beyond a size, e.g., six, their movement halts. Afterward, evolution is slow and growth depends on single atoms diffusing nearby and attach. This also explains that small clusters or even single atoms could remain highly dispersed once synthesized at loadings as high as 3 wt%[28,29]. The co-existence of single atoms and small clusters has also been observed for $Pd_n$ on $In_2O_3$[39]. CO adsorption locks the cluster growth, reduces the sizes, narrows down the size distribution ($\sigma_n$ ~1.6). Higher loadings lead to larger clusters. Experiments have shown both dispersed single atoms and clusters on $CeO_2$[29]. Clusters of 0.5–1.2 nm have been reported under CO. Examples include $Pd_n$ on ceria[40], $Pt_n$ on ceria[26], $Pd_n$ on $Fe_3O_4$[41], and $Pt_n$ on $TiO_2$[42]. In CO oxidation, intermediate-sized clusters have been reported to be more active than single atoms, dimers, trimers, or larger nanoparticles[43]. Finally, Gänzler et al. suggested 1.4 nm as optimal for $Pt_n$ on $CeO_2$[44]. Our results agree qualitatively with experiments about intermediate sizes being dominant.

In these simulations, error arises from the DFT calculations and the ML Hamiltonians. The Pd and CO Hamiltonians have a testing RMSE (root mean square error) of 1.01 meV/site and 0.23 eV/site, respectively. We account for key temperature effects, using classical statistical mechanics and collision theory to estimate rate constants for Pd diffusion, CO adsorption, and CO desorption, and the contribution of configurational entropy on the relative stabilities of supported structures. The vibrational entropy of Pd atoms and anharmonic effects are not considered and are expected to play some role at higher temperatures. Other methods, such as the NASA polynomials[45], could be used instead to account for temperature effects. In our experience, the statistical error of KMC simulations is rather small and can be reduced by performing calculations in parallel with different seeds of the random generator. To simplify the model, we leave out Pd-O interactions[45], such as O vacancy formation[46,47]. A strength of the model is that it can easily be extended to account for such effects and binding sites, such as "atom trapped" single atoms[6,29] or steps at the $CeO_2$ surface[47] by including the relevant diffusion events and their barriers estimated using DFT. Future work should extend the framework to a broader range of working conditions and compare to dedicated experimental data.

In summary, this paper investigates the dynamics of subnanometer clusters in a vacuum and a CO pressure at room and elevated temperatures. The computational framework integrates DFT calculations, machine-learning-enable multiscale modeling, and graph-theoretical KMC simulations. The approach offers insights into the dynamics of cluster formation at experimentally relevant time scales. Sintering happens even at room temperature driven by the reduction in free energy. It occurs via single atom and small cluster diffusion. It leads to a distribution of sizes and shapes (clusters with less than 20 atoms—a diameter <1.2 nm) in minutes due to sluggish kinetics. CO adsorption happens at a comparable rate to 3D cluster growth at room temperature. It is

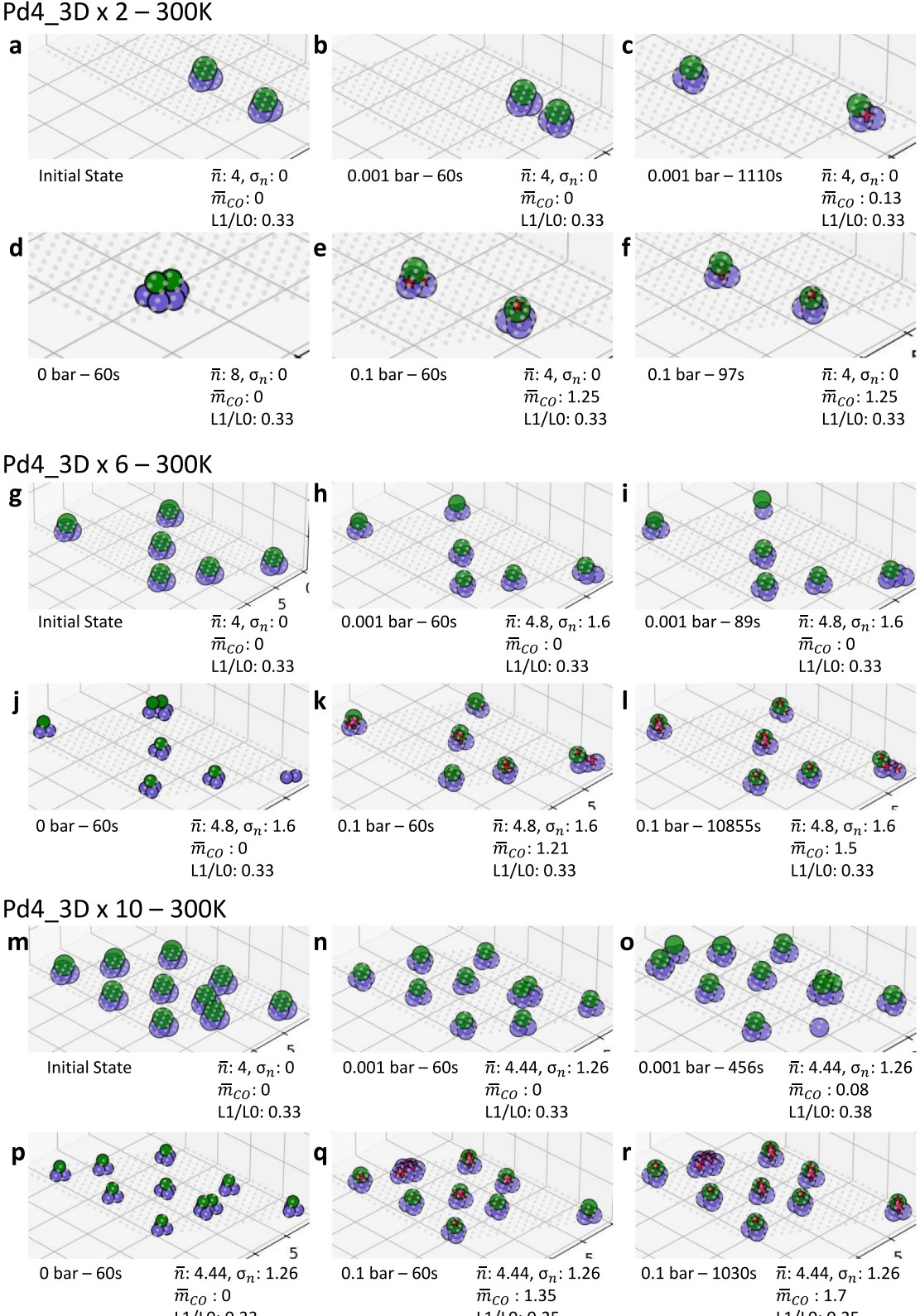

**Fig. 5 Effect of pre-sintering and CO partial pressure on catalyst dynamics at 300 K.** Initial state of (**a–f**) 2, (**g–l**) 6, (**m–r**) 10 Pd$_4$_3d clusters at 300 K. (coverages of 0.02, 0.06, and 0.1 or 0.23, 0.69, and 1.15 wt%).

slower than small bare cluster diffusion, and thus, it affects more extended time scale dynamics and structures. It leads to flatter structures and smaller sizes than bare Pd metal to maximize the bridge and hollow sites near the metal/oxide interface where CO adsorption is stronger. The structures seen with or without CO at room temperature are kinetically frozen and differ from their thermodynamic (equilibrium) ones. IR and chemisorption experiments at near room temperature will probe these kinetically frozen, tiny, two-layer structures using CO as a probe molecule and large clusters upon annealing at high temperatures in He. Thermodynamic stability (the long-time behavior) and associated calculations, while useful, could be irrelevant. Elevated temperatures lead to stable or metastable structures rapidly (in seconds), consistent with thermodynamic calculations due to sufficient thermal energy. Upon gradual heating from room temperature, e.g., in a temperature-programmed desorption mode, the experimental IR spectra could probe simultaneously the dynamics of further sintering along with changes induced by the CO desorption—the time scales would be attainable. Our results highlight the importance of kinetics and time scale information for experimental characterization and catalyst design and that low-temperature characterization data may be less relevant to the real-time high-temperature dynamics.

## Methods

**Energetics calculations**. The setting of DFT calculations is provided in Supplementary Note 1. The details of ML-based Hamiltonian energy models and the lattice representation are described in Supplementary Note 2 and 3, respectively.

**Kinetic Monte Carlo Simulations**. We input the ML-predicted energetics and calculated vibrational frequencies along with the elementary dynamic events with initial and final states represented as lattice graphs to the KMC Zacros software[48]. Each pattern in the Hamiltonians is a subgraph of the lattice. Zacros implements an efficient rejection-free algorithm with local propensity updates (microscopic rates). The framework advances the time clock by adding an interarrival time between events. Supplementary Note 4 provides an overview and more details are given in review papers[49,50]. Examples of its applications in atomistic ripening are in the references[9,22,51,52].

For the dynamics, we consider (1) bare Pd single atoms and clusters (partial CO pressure of 0 bar) and (2) their counterparts exposed to CO on a $CeO_2(111)$ support. By turning on the partial pressure $P_{CO}$, we explore the catalyst dynamics upon exposure to CO, as in IR experiments. We include diffusion of Pd single atoms and small clusters ($n \leq 4$) and CO adsorption and desorption when $P_{CO} > 0$. Larger clusters are stationary due to high diffusion barriers (Supplementary Table 2). CO diffusion on $Pd_n$ clusters is ignored due to the small cluster size and the mobility based on desorption and readsorption through the gas-phase. All microscopic events are reversible. We estimate the partition functions of adsorption/desorption events from the vibrational frequencies and other thermodynamic properties using pMuTT[53]. We approximate the pre-exponential factors (prefactors) of $Pd_n$ diffusion as $10^{13}\,s^{-1}$ ($\approx \frac{k_B T}{h}$). We provide a full catalogue of microscopic events and parameters in Supplementary Note 5, 6. The adsorption/desorption and diffusion events have very dissimilar rates that makes a KMC simulations sample only fast quasi-equilibrated events[32]. A suitable downscaling of the rate constants of quasi-equilibrated events does not change the simulation outcome[33–35]. Here, we auto-scale the pre-exponential factors of fast events in Zacros and analyze simulation results using the Wrapper[54]. Details of the KMC acceleration are in Supplementary Note 7.

## Data availability

Data of this study are available within the paper and its Supplementary Information. Supplementary Information contains: (1) Additional simulation results; (2) details of the computational methods, including all elementary events and model parameters. Data of the species counts vs. time data, additional plots, and animations produced from the Zacros output files are available on GitHub at https://github.com/VlachosGroup/Pdn-CO-Dynamics [55]. The raw Zacros output files are available from the corresponding author upon request. Supplementary Movie 1 and Supplementary Movie 2 show the lattice animations for 125 Pd single atoms on a 50 by 50 lattice (coverage = 0.05) in a vacuum and a CO pressure of 0.1 bar, respectively. Supplementary Movie 3 and Supplementary Movie 4 show the lattice animations for 20 Pd single atoms on a 20 by 20 lattice (coverage = 0.05) in a vacuum and a CO pressure of 0.1 bar, respectively. Supplementary Movie 5 and Supplementary Movie 6 show the lattice animations for 6 Pd4_3d clusters on a 20 by 20 lattice (coverage = 0.06) in a vacuum and a CO pressure if

0.1 bar, respectively. Supplementary Movie 7 shows the lattice animation for a $Pd_{20}$ cluster in a vacuum. All Supplementary Movies are stimluated at 300 K.

## Code availability

Sample Zacros input files are available in the same GitHub repository at https://github.com/VlachosGroup/Pdn-CO-Dynamics [55]. The directory structure is described in the readme file of the repository.

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

## Acknowledgements

The work of Y.W., J.K. and D.G.V. was supported as part of the Catalysis Center for Energy Innovation (CCEI), an Energy Frontier Research Center funded by the US Dept. of Energy, Office of Science, Office of Basic Energy Sciences under award number DE-SC0001004. CCEI has the main intellectual contribution to the manuscript. E.J.M.H. and Y.Q.S. acknowledge the financial support from The Netherlands Organization for Scientific Research (NWO) through a Vici grant. Supercomputing facilities were provided by NWO and Hefei Advanced Computing Center and from the European Union's Horizon 2020 research and innovation programme under grant No 686086 (Partial-PGMs). Y.Q.S. acknowledge the "Young Talent Support Plan" Fellowship of Xi'an Jiaotong University. The authors acknowledge Xue Zong for reviewing and improving the manuscript.

## Author contributions

J.K. contributed to the KMC simulations and data analysis. Y.Q.S. and Y.W. contributed to the DFT calculations, machine learning and the rest of the modeling framework. E.J.M.H. and D.G.V. provided supervision. The manuscript was written by Y.W. and D.G.V. All authors contributed to analyzing the results and commenting on the manuscript.

## Competing interests

The authors declare no competing interests.
