## [Peer Review File · Nature Communications]

Real-time dynamics and structures of supported subnanometer catalysts via multiscale simulationsREVIEWER COMMENTS

Reviewer #1 (Remarks to the Author):

The authors employed a combination of theoretical approaches to study the dynamics and structures of CeO₂-supported Pd clusters. My main concern is the novelty of this work. While modern approaches like machine learning Hamiltonians were used, no new chemistry/catalysis was found. Dynamics and structures of supported metal clusters have been routinely investigated using DFT-based KMC modeling (there are many examples including Xu et al., Surf. Sci. 601, 3133, 2007). I also think this manuscript is too technical and better suited to a more specialized journal.

Regardless of where this manuscript is published, it might help the reader by considering the following:

1. If I understood correctly some the DFT energies and ML energies in this work are from Refs 3 and 4 in the SI. While in these Refs, Pd structures with n=1-21 were investigated in this work consider clusters with n=1-40 were investigated. Please provide missing data. Also, the error of ML Hamiltonian models should be provided. Furthermore, while all the reference energies were calculated at 0 K. I wonder if the ML models are reliable to present the free energy surface at 700 K.
2. Lines 114-115: CO adsorption slows down Pd diffusion and the overall dynamics. Intuitively, CO adsorption weakens Pd-CeO₂ interactions, making Pd diffusion faster. I would double check this with DFT calculations for diffusion barriers.
3. Line 233, can a cluster with less than 40 atoms have a 1.4 nm diameter?
4. Line 279, the authors used the pre-exponential factors of diffusion as 10¹³. In fact, the pre-factor of CO diffusion on solids strongly depends the coverage (then, CO pressure) (it can be changed by few orders, see for example, Hulva et al., J. Phys. Chem. B 2018, 122, 2, 721–729).

Reviewer #2 (Remarks to the Author):

The paper "Real-time Dynamics and Structures of Supported Subnanometer Catalysts via Multiscale Simulations" by Yifan Wang et al., presents a multiscale simulation study of Pd clusters supported on CeO₂ (111) surface, also in the presence of CO. The study is a breakthrough in understanding supported metal catalysts, and is a conclusion of a large chunk of work done by the authors over several years. From this point of view, the work can be considered for publication in Nature Comm.. However, there are also several issues remaining that I think were not addressed properly by the authors. Without discussing these issues it remains unclear how the fascinatingly complex model developed by the authors is related to realistic catalysts. My detailed comments are below.

Major comments:

- 1) The authors have used PBE+U for their calculations. However, this DFT functional does not include the long-range van der Waals interaction, which may be important for adsorbed metallic clusters. The authors should justify this choice.
- 2) The authors should justify their model simplifications. In particular, the authors should discuss:
 - steps at CeO₂ surface; steps can have a strong influence on Pd dispersion (see e.g. DOI 10.1038/ncomms10801);
 - defects; the authors did not consider processes of O vacancy formation, which can have a strong effect on Pd cluster structure and interaction with CO;
 - Pd cluster oxidation/reduction by support;

-vibrational dynamics; the authors consider system in thermal equilibrium; however, local changes in temperature of the Pd clusters upon Pd atom-cluster attachment/detachment, and CO adsorption/desorption, may have a longer equilibration time than the considered processes.

3) To make the paper more accessible to general readership of Nature Comm., the authors should clearly describe the on-going discussions in the research area (with the larger scope of supported metal catalysts in general) and point out the plethora of problems preventing the field from moving forward at a better pace.

Minor comments:

"The rates of elementary events reveal the critical steps and are illuminating ()." - empty parentheses?

"Upon pre-factor rescaling" - explain what this is for a broad audience

"CO adsorption and uphill Pd1 hopping" - clarify what "uphill" means

The authors should explain why they do not consider adsorption of CO at a bare CeO₂(111) surface.

"We show the time evolution of the various complexes (normalized by Pd atom, n)" - unclear what is meant by "(normalized by Pd atom, n)"

"The CO chemisorption-seen stoichiometry" - clarify what this is

"Unexpectedly, it is unlikely to sinter single atoms within minutes into large nanoparticles at room temperature." - likely or unlikely? unlikely would not be unexpected

Reviewer #3 (Remarks to the Author):

Wang, Kalscheur, Su, Hensen, and Vlachos simulated the reconstruction/nucleation of Pd nanoclusters supported on CeO₂(111) under various Pd stoichiometries, temperatures, and by adding CO. Due to the large span of the time scales, they produced a Multiscale model by combining well-known Kinetic Monte Carlo and Density Functional Theory with relatively novel cluster expansion with active learning, and a cluster genetic algorithm. They have found that, even at low temperatures, the Pd clusters sinter in seconds. Such a time scale is relatively long for simulations containing atomistic information. At room temperatures, the clusters become trapped in far-from-equilibrium states once the clusters reached a certain size, as the diffusion is less expensive in small clusters. The sintering is exacerbated at higher temperatures, as the material gains enough energy to be released from its thermodynamic trap. The material tends to be "flatter", arraying as bilayers in the presence of CO.

Due to the elegant combination of techniques, I have found the study highly appealing for the computational modeling community. Yet, it can be made a lot more appealing after the minor revisions noted below. The wording was sometimes hard to follow. I had to read twice many sentences to understand their meaning, even when they explained simple facts and concepts. Few examples below:

* Figure 1(b) The lattice of Pd_n resembles one layer of a simple-cubic packed metal, instead of a closed-packed. What is going on there?

* Figure 1(c) the seven colors assigned for bridge and hollow positions are indistinguishable. They also cannot be clearly distinguished by height. CO-Top is very hard (but not impossible) to distinguish, although it helps that its symbol is a star.

* Many phrases are built with an unnecessarily high level of abstraction. For instance "Labeled Pd sites and CO sites", line 81, would be a lot more understandable (in one read) if the authors focus a bit on the physical phenomenon being modeled: "Labeled Pd sites and CO *adsorption* sites".

* The videos need an extensive revision. They are shown with many digits that are not significant; the units abbreviations are wrong; eg, "11.399999999999991 sec" would better read "11.4 s". "ang" would better be replaced by "Å". Now about cosmetics: As it is right now, less than 20% of the space of the canvas contains useful information, while the rest is noise. The grey box hinders a lot of the visibility of the rhombohedral cell and the observation of diffusion. It would be better to show 1 unit cell, well-marked, and perhaps extend in other directions. Also to halve the size of y-coord and greatly reduce the z-coord. For instance, the panels shown in Figure 4 do a much better use of the space. All previous comments also apply to the GitHub repository.

* The author's notation is unnecessarily confusing. For instance, Pd1, Pd2, Pd3 in Figure 2 (and elsewhere) are written without subscripts but yet referring to a monomer, a dimer, and a trimer, respectively. Why not use subscripts when such numbers refer to stoichiometry? Then, notations are sometimes not consistent, eg Pd4-3D and Pd4_3D, lines 201,205,207.

* Figure 2 and most Supp. Figures are missing units! Is the pressure in bar or mbar? Does " ξ " have units? It is not evident at a first sight what the caption "time evolution" is referring to without an intensive revision of all methods. How long was the sampling time for the "event frequencies"?

* Line 112.

* The authors jump too fast to the results and conclusions without first explaining how the pieces couple to each other. Only a mention to the Methods in line 90. Then the Methods section (L256-284) explains more what the authors did but not much about how the pieces fit together. They just refer the reader to Supplementary Figure 1 in line 256. Supplementary Figure 1 is actually the most important in all this work whose novelty is, precisely, the methods being used in this particular sequence. At least one paragraph detailing how the pieces fit together is needed when introducing the Results section. This is, *not* what the authors did, which is already well explained, but rather *why* they did it in that order, in few words.

Other minor points:

* The observation "single atoms do not disappear after agglomeration but always co-exist with clusters" is not fully new. It has also been observed for Pd on In₂O₃ (Nat Commun, 2019, 10, 3377). There, intermediate-size clusters are also active for CO₂ reduction to CO, whereas low-nuclearity ones are more selective to methanol.

* The conclusion "low-temperature characterization data provides little insights into high-temperature catalysis" is perhaps an overstatement. A catalyst exposed to high temperatures, once major sintering/reconstruction has occurred, can then be characterized at room temperatures without large deviations. This is particularly true for single-atom and "extended" catalysts. The authors would better rephrase or, alternatively, show that the catalysts do change their behavior after annealing from high to room temperature.

Summary of Overall Changes in Response to the Reviewer Comments

- We reviewed the challenges in the research of supported metal catalysts and provided an overview of the developed computational framework and its rationale in the results' session.
- We resolved all the formatting and phrasing issues to make the paper more accessible to the broad readership of Nature Communications.
- We revised the videos (Supplementary Movie 1-7) and figures in the manuscript, SI, and GitHub repository according to the reviewers' suggestions. We improved the overall clarity of figures, captions, and notations. The descriptions of the Supplementary Movies can be found in the Data availability section in the manuscript.
- We provided detailed discussions on our approach's limitations and the error of each computational method.
- We rewrote key messages to make them more general to appeal to even a broader readership following the comment of Reviewer 1.

Point by Point Comments and Responses to Reviewers' Comments

Red text indicates our response and action, blue text is our edited or new text, black text is the reviewers' comments. The changes in the manuscript and SI are highlighted in yellow.

Reviewer #1

The authors employed a combination of theoretical approaches to study the dynamics and structures of CeO₂-supported Pd clusters. My main concern is the novelty of this work. While modern approaches like machine learning Hamiltonians were used, no new chemistry/catalysis was found. Dynamics and structures of supported metal clusters have been routinely investigated using DFT-based KMC modeling (there are many examples including Xu et al., Surf. Sci. 601, 3133, 2007). I also think this manuscript is too technical and better suited to a more specialized journal.

Response: This is a good point to clarify. We acknowledge that our work does not cover any new chemistry and DFT-based KMC has been applied to study growth of clusters on Mg in a previous study. Specifically, we cited Xu et al. in reference 22. Our work differs from prior work on several fronts. Methodologically, ML Hamiltonians-based KMC models cluster structures have much higher flexibility and computational efficiency; we employ a two Hamiltonian approach that is additive to allow consistent modeling not just of the metal but of adsorbates in the zero coverage limit and also for high coverages. Compared to prior work, our work focuses on three-dimensional (3D) representations of metal clusters, the effect of CO adsorption on cluster dynamics that is essential to catalysis compared to growth, and long-time scales of relevance to experiments. These efforts allow us to simulate real experimental characterization (such as IR) conditions. Besides from the methodology, the novelty lies in its near experimental settings and the addition of gas adsorbates to the system that open up the door for spectroscopic and catalysis studies.

Comment 1

Regardless of where this manuscript is published, it might help the reader by considering the following:
1. If I understood correctly some the DFT energies and ML energies in this work are from Refs 3 and 4 in the SI. While in these Refs, Pd_n structures with n=1-21 were investigated in this works consider clusters with n=1-40 were investigated. Please provide missing data. Also, the error of ML Hamiltonian models should be provided.

Response 1

We thank the reviewer for the suggestions. As it was mentioned in the first paragraph of the Results section: “details of all sub-models can be found in Supplementary Methods and prior works^{10,23}”, we provide the machine learning model energetics (for both Pd_n and CO on Pd_n) related to this work in the SI and point the readers to references 10 and 23 for the more detailed discussions on the Pd_n model building and DFT dataset. The DFT dataset used for training the Pd_n model contains the energies and structures of Pd_n structures with n=1-21. In reference 23 (DOI: 10.1021/acsnano.0c06472), we show that the model can be extrapolated to clusters with size greater than 21 and up to ~50 we tested then. The same model is used in this work as the Pd_n Hamiltonian. There is no missing data, and the choice of machine learning energetics can be justified. We expand our discussion on model uncertainty and summarize the error on page 9.

We added a statement to clarify that all data for the Pd_n model was provided (Page 11):
All data for the Pd_n model was provided and validated in Ref. 23.

We added the following text:

In these simulations, error arises from the DFT calculations and the ML Hamiltonians. The Pd and CO Hamiltonians have a testing RMSE (root mean square error) of 1.01 meV/site and 0.23 eV/site, respectively. Furthermore, we are mainly interested in the contribution from configurational entropy and the relative stabilities of supported structures. We use classical statistical mechanics and collision theory

to estimate rate constants for Pd diffusion, CO adsorption, and CO desorption. Vibrational entropy of Pd atoms could become important at higher temperatures. Anharmonic effects are not considered and are expected to play some role at higher temperatures. Other methods, such as the NASA polynomials,⁴⁷ could be used instead to account for temperature effects. The statistical error of KMC simulations is rather small and can be reduced by performing parallel calculations with different seeds of the random generator. To simplify the modeling effort, we leave out Pd-O interactions⁴⁸, such as O vacancy formation.^{49,50} Finally, the model can easily be extended to account for binding sites such as "atom trapped" single atoms^{28,30} or steps at the CeO₂ surface⁵⁰ by including such sites and the relevant diffusion events once their barriers are computed using DFT. Future work should extend the framework to a broader range of working conditions and compare to dedicated experimental data.

Comment 2

Furthermore, while all the reference energies were calculated at 0 K. I wonder if the ML models are reliable to present the free energy surface at 700 K.

Response 2

Thanks for this insightful comment. The ML models are trained on the DFT energies at 0 K. Statistical mechanics is used to introduce temperature effects. The KMC framework introduces temperature and time dependence by computing the propensities (or rates) of key events. Every occurrence of a key event indicates that the system jumps from one state to another and therefore we can compare their relative stability at a certain temperature. For this work, we are mainly interested in the contribution of configurational entropy for metal atoms, which is captured by the Pd diffusion and hopping events. The vibrational frequency contribution is generally small for solid-state systems and almost similar for various isomers on a support (DOI: 10.1103/PhysRevB.67.193407, 10.1002/jcc.24379). For CO adsorption and desorption, vibrational and translational contributions are considered explicitly in the rate calculations. More details can be found in the KMC literature, for example, reference 51-53. We acknowledge that the error could become significant at higher temperatures such as 700 K. We have expanded on this point in the revised manuscript on page 9-10 (see our reply to comment 1).

Comment 3

2. Lines 114-115: CO adsorption slows down Pd diffusion and the overall dynamics. Intuitively, CO adsorption weakens Pd-CeO₂ interactions, making Pd diffusion faster. I would double check this with DFT calculations for diffusion barriers.

Response 3

Thank you for pointing this out. We double-checked our results. From DFT calculations, the diffusion barriers are higher for CO attached on small clusters compared to the bare clusters with $n=1-3$ (Pd₁ - 0.14 eV vs. Pd₁(CO) - 0.62 eV, Pd₂ - 0.37 eV vs. Pd₂(CO) - 0.52 eV, Pd₃ - 0.71 eV vs. Pd₃(CO) - 1.17 eV). The barriers are shown in the Supplementary tables 2 and 3 of the SI (shown below). The event frequencies and timescales of key events in Figure 2 also suggest similar findings. As suggested in reference 10 (DOI: 10.1021/acs.chemmater.7b03555), CO binds strongly to single atoms and small clusters ($n=1-3$) and the weakening effect of metal-support interaction might not be apparent. However, for larger two-layer clusters such as Pd₄ 3d, the diffusion barrier is higher for the bare cluster (1.35 eV) compared to the CO attached cluster (1.03 eV). CO adsorption makes their diffusion faster. The effect of CO on diffusion appears to depend on the size and structure of the clusters. In our system, the main driver for diffusion are the small clusters with $n=1-3$, and therefore we conclude that CO adsorption slows down the Pd diffusion and the overall dynamics of the smaller clusters but has an opposite effect for larger clusters. We have extended our discussion and clarified our discussion in the revised manuscript. (Page 5)

Small CO-Pd cluster diffusion is also rapid but slower than bare cluster diffusion (\$n \leq 3\$ ), i.e., CO adsorption slows down the Pd diffusion and the overall dynamics of small clusters (the barriers are shown

in Supplementary Tables 2 and 3). However, for larger two-layer clusters, such as Pd₄_3d, the diffusion barrier for the bare cluster (1.35 eV) is higher compared to the one when CO is attached to the cluster (1.03 eV). CO adsorption makes their diffusion faster.

Supplementary Table 2. List of the elementary events included in the bare Pd_n system.

Index	Elementary event	Barrier E_a (e)	Prefactor A_{fwd} (s^{-1})	A_{ratio}
1	Pd ₁ diffusion on layer 0	0.14	1.00E+13	1
2	Pd ₁ diffusion on layer 1	0.05	1.00E+13	1
3	Pd ₁ diffusion on layer 2	0.05	1.00E+13	1
4	Pd ₁ diffusion on layer 3	0.05	1.00E+13	1
5	Pd ₂ diffusion on layer 0	0.37	1.00E+13	1
6	Pd ₃ diffusion on layer 0	0.71	1.00E+13	1
7	Pd ₄ diffusion on layer 0	1.35	1.00E+13	1
8	Pd ₄ _3d diffusion on layer 0	1.35	1.00E+13	1
9	Pd ₁ hopping from layer 0 to layer 1	0.85	1.00E+13	1
10	Pd ₁ hopping from layer 1 to layer 2	0.85	1.00E+13	1
11	Pd ₁ hopping from layer 2 to layer 3	0.85	1.00E+13	1

Supplementary Table 3. List of additional elementary events included in the Pd-CO system.

Index	Elementary event	Barrier E_a (e)	Prefactor A_{fwd} (s^{-1})	A_{ratio}
12-20	CO adsorption on various sites	0	9.67E-02	5.20E-14
21	Pd ₁ (CO) diffusion on layer 0	0.62	1.00E+13	1
22	Pd ₂ (CO) diffusion on layer 0	0.52	1.00E+13	1
23	Pd ₃ (CO) diffusion on layer 0	1.17	1.00E+13	1
24	Pd ₄ _3d(CO) diffusion on layer 0	1.03	1.00E+13	1

Comment 4

3. Line 233, can a cluster with less than 40 atoms have a 1.4 nm diameter?

Response 4

From our model, a cluster of size n=38 has the diameter of 1.27 nm. We would expect the cluster size in reference 35 to contain more than 40 atoms. In the manuscript, by citing this work, our objective is to show that our results agree qualitatively with the experiments, namely, the stable cluster size at long times does not further increase beyond 1.4 nm at the given conditions.

Comment 5

4. Line 279, the authors used the pre-exponential factors of diffusion as 10¹³. In fact, the pre-factor of CO diffusion on solids strongly depends the coverage (then, CO pressure) (it can be changed by few orders, see for example, Hulva et al., J. Phys. Chem. B 2018, 122, 2, 721–729).

Response 5

Thank you for pointing us towards the reference and apologies for the confusion. We assume the pre-exponential factors of diffusion for Pd atoms on ceria support as 10¹³, which should be less dependent on the coverage and pressure. For CO, the pre-exponential of adsorption and desorption were obtained from DFT (Supplementary Table 3). The CO diffusion was ignored in our model due to the small sizes of the

clusters and the possibility of CO to move around due to adsorption and desorption. We modified the text on page 12:

CO diffusion on Pd_n clusters is ignored due to the small cluster size.

We approximate the pre-exponential factors (prefactors) of Pd_n diffusion as 10^{13} s^{-1} ($\approx \frac{k_B T}{h}$).

Reviewer #2

The paper "Real-time Dynamics and Structures of Supported Subnanometer Catalysts via Multiscale Simulations" by Yifan Wang et al., presents a multiscale simulation study of Pd clusters supported on CeO₂ (111) surface, also in the presence of CO. The study is a breakthrough in understanding supported metal catalysts, and is a conclusion of a large chunk of work done by the authors over several years. From this point of view, the work can be considered for publication in Nature Comm.. However, there are also several issues remaining that I think were not addressed properly by the authors. Without discussing these issues it remains unclear how the fascinatingly complex model developed by the authors is related to realistic catalysts. My detailed comments are below.

Response: We appreciate the recommendation for publication.

Comment 1

1) The authors have used PBE+U for their calculations. However, this DFT functional does not include the long-range van der Waals interaction, which may be important for adsorbed metallic clusters. The authors should justify this choice.

Response 1

We considered the effect of the van der Waals interactions by setting the IVDW = 12 in the INCAR for all the DFT calculations we performed. When setting IVDW=12, the DFT-D3 method with the Becke-Jonson damping is employed. We added the justification into the SI (Page 10).

The long-range van der Waals interactions are included by setting the IVDW in the INCAR for all the DFT calculations to 12 (the DFT-D3 method with the Becke-Jonson damping is employed).

Comment 2

2) The authors should justify their model simplifications. In particular, the authors should discuss:
-steps at CeO₂ surface; steps can have a strong influence on Pd dispersion (see e.g. DOI 10.1038/ncomms10801);
-defects; the authors did not consider processes of O vacancy formation, which can have a strong effect on Pd cluster structure and interaction with CO;
-Pd cluster oxidation/reduction by support;
-vibrational dynamics; the authors consider system in thermal equilibrium; however, local changes in temperature of the Pd clusters upon Pd atom-cluster attachment/detachment, and CO adsorption/desorption, may have a longer equilibration time than the considered processes.

Response 2

These are excellent points. Since the vibrational contribution is generally small and almost similar for various isomers on the support at low temperatures (DOI: 10.1103/PhysRevB.67.193407, 10.1002/jcc.24379), we do not include its contribution for Pd atoms. For CO adsorption and desorption, we consider the vibrational and translational contributions explicitly in KMC using statistical mechanics. We did not include these discussions initially due to the word limit. In the revised manuscript, we added a paragraph to explain the model and its limitations further (pages 10-11).

See our reply to Reviewer 1, comment 1.

Comment 3

3) To make the paper more accessible to general readership of Nature Comm., the authors should clearly describe the on-going discussions in the research area (with the larger scope of supported metal catalysts

in general) and point out the plethora of problems preventing the field from moving forward at a better pace.

Response 3

Thank you for raising this point. We improved the literature review section to connect with a broader picture and pointed out specific challenges in the field of subnanometer catalysts (Page 2):

Supported platinum group metal (PGM) catalysts are used in the catalytic converter of vehicles to convert exhaust gases. The development of low-cost, high-performing catalysts will continue for achieving emission standards.¹ Supported single-atom catalysts (SACs) and subnanometer clusters offer unique catalytic properties for several industrial relevant chemistries,²⁻⁵ and their high dispersion (size n , $n \sim \leq 30$) exposes nearly all metal atoms for catalysis. Great efforts in recent literature are devoted to developing thermally stable, atomically dispersed SACs and subnanometer cluster catalysts.⁶ Loss of dispersion and catalyst deactivation could occur over time due to sintering via (1) individual particle migration and coalescence and (2) Ostwald ripening, where single atoms detach from a particle, diffuse on the support, and attach to another particle.^{9,10} Sintering depends strongly on metal loadings and environmental factors, such as temperature, atmosphere, and metal-support interactions.¹¹ To maintain dispersion, metal loadings are typically very low, and this makes *in-situ* and *operando* characterization very challenging. Aside from the detection limit, the time resolution of experimental methods is medium to long and this precludes insights into fast catalyst dynamics.^{1,8,12} Multiple experimental techniques are used to characterize the catalysts but their effect on the catalyst structure and dynamics, e.g., upon exposure the metal to CO for IR measurements, is unknown. The catalyst activity and selectivity are dictated in part from their structure controlled by metal-metal, metal-adsorbate, and metal-support interactions.^{7,8} Mastering the structure evolution at the atomic scale under working conditions is key to understanding structure-reactivity relations and discovering new catalysts.

Comment 4

"The rates of elementary events reveal the critical steps and are illuminating ()." - empty parentheses?

Response 4

It should be Figure 2 inside the parentheses.

The rates of elementary events reveal the critical steps and are illuminating (Figure 2). (Page 5)

Comment 5

"Upon pre-factor rescaling" - explain what this is for a broad audience

Response 5

We explained what pre-factor rescaling is and pointed the reader to Supplementary Methods 7 for more details.

Pre-factor rescaling is a common multiscale technique in KMC to advance the time clock to long times²⁷⁻³⁰ without impacting the results (Supplementary Methods 7). (Page 5)

Comment 6

"CO adsorption and uphill Pd₁ hopping" - clarify what "uphill" means

Response 6

Sorry for the confusion. We were referring to the large energy barrier of Pd₁ hopping. We modified the text as below:

CO adsorption and Pd₁ hopping from layer 0 to layer 1 are the slowest, with a time scale between milliseconds (10^{-3} s) and seconds. The latter must overcome a large energy barrier. (Page 5)

Comment 7

The authors should explain why they do not consider adsorption of CO at a bare CeO₂(111) surface.

Response 7

Thank you for raising this point. Our DFT calculations show that CO adsorption on the bare CeO₂(111) is weak with an adsorption energy of -0.20 eV. The distance between a CO and the CeO₂(111) surface is 2.87 Å, indicating a typical physisorption. Therefore, we did not take it into account. We addressed this point in the SI (Page 16).

It should be noted that only CO chemisorption on Pd atoms is considered. Our DFT calculations show that CO adsorption on the bare CeO₂(111) is weak with an adsorption energy of -0.20 eV. The distance between a CO and the CeO₂(111) surface is 2.87 Å (Supplementary Fig. 14), indicating a typical physisorption. Therefore, we did not include it into the model.

Supplementary Figure 14. CO physisorption on the bare CeO₂(111) surface.

Comment 8

"We show the time evolution of the various complexes (normalized by Pd atom, n)" - unclear what is meant by "(normalized by Pd atom, n)"

Response 8

Thanks for pointing this out. We meant divided by the number of Pd atoms as a function of time. We rephrased the sentence to (Page 6)

We show the time evolution of the various species counts (divided by the number of Pd atoms, n) in Figure 3.

Comment 9

"The CO chemisorption-seen stoichiometry" - clarify what this is

Response 9

We modified the text to (Page 7)

The CO chemisorption stoichiometry (\bar{m}_{CO})

We have defined the stoichiometry in the previous paragraph (Page 6):

The number of CO molecules per Pd atom \bar{m}_{CO} , also known as stoichiometry, is a good descriptor for CO adsorption.

Comment 10

"Unexpectedly, it is unlikely to sinter single atoms within minutes into large nanoparticles at room temperature." - likely or unlikely? unlikely would not be unexpected

Response 10

It should be unlikely. We were trying to show that the sintering stops before large nanoparticles were formed. We removed the word “unexpectedly” to avoid confusion (Page 7).

It is unlikely to sinter single atoms within minutes into large nanoparticles at room temperature.

Reviewer #3

Wang, Kalscheur, Su, Hensen, and Vlachos simulated the reconstruction/nucleation of Pd nanoclusters supported on CeO₂(111) under various Pd stoichiometries, temperatures, and by adding CO. Due to the large span of the time scales, they produced a Multiscale model by combining well-known Kinetic Monte Carlo and Density Functional Theory with relatively novel cluster expansion with active learning, and a cluster genetic algorithm. They have found that, even at low temperatures, the Pd clusters sinter in seconds. Such a time scale is relatively long for simulations containing atomistic information. At room temperatures, the clusters become trapped in far-from-equilibrium states once the clusters reached a certain size, as the diffusion is less expensive in small clusters. The sintering is exacerbated at higher temperatures, as the material gains enough energy to be released from its thermodynamic trap. The material tends to be "flatter", arraying as bilayers in the presence of CO.

Comment 1

Due to the elegant combination of techniques, I have found the study highly appealing for the computational modeling community. Yet, it can be made a lot more appealing after the minor revisions noted below. The wording was sometimes hard to follow. I had to read twice many sentences to understand their meaning, even when they explained simple facts and concepts. Few examples below:

Response 1:

We appreciate the positive feedback from the reviewer, and we apologize for the confusion caused.

Comment 2

* Figure 1(b) The lattice of Pd_n resembles one layer of a simple-cubic packed metal, instead of a closed-packed. What is going on there?

Response 2

We assume the structure is composed of alternating planes of closest-packed spheres (ABCABCABC). Many metals, including Ag, Al, Au, Ca, Co, Cu, Ni, Pb, and Pt, crystallize in such structure. For more details, please see reference 23 and this post. The same lattice is used for the Pd_n clusters in this work. We have attached Figure S1 from reference 23 (DOI: 10.1021/acsnano.0c06472) below:

Editorial Note: Reprinted with permission from Wang, et al., Finite-Temperature Structures of Supported Subnanometer Catalysts Inferred via Statistical Learning and Genetic Algorithm-Based Optimization, *ACS Nano* 2020 14 (10), 13995-14007. Copyright 2020 American Chemical Society.

Figure S1. Pd lattice representation and unit cell. (a) FCC lattice showing A, B, and C sites. (b) 2D projection of a unit cell with the neighboring A sites included. (c) A unit cell for the 3D lattice with nearest neighboring connections and corresponding site types and layer numbers labelled.

To be more specific, we modified the text to

In the case of bare Pd_n, the Pd lattice consists of 4 closest-packed layers above the support. (Page 10 in the SI)

Comment 3

* Figure 1(c) the seven colors assigned for bridge and hollow positions are indistinguishable. They also cannot be clearly distinguished by height. CO-Top is very hard (but not impossible) to distinguish, although it helps that its symbol is a star.

Response 3

Thank you for raising this point. We modified the colors in the original Figure 1(b) and (c) and added the computational framework to Figure 1 (see Comment 8).

Figure 1. Modeling framework for subnanometer catalysts using first-principles multiscale modeling and machine learning (ML) on the lattices of ceria supported Pd subnanometer clusters.

Comment 4

* Many phrases are built with an unnecessarily high level of abstraction. For instance "Labeled Pd sites and CO sites", line 81, would be a lot more understandable (in one read) if the authors focus a bit on the physical phenomenon being modeled: "Labeled Pd sites and CO *adsorption* sites".

Response 4

Thank you for pointing this out. We changed "CO sites" to "CO adsorption sites" throughout the paper and SI.

Comment 5

* The videos need an extensive revision. They are shown with many digits that are not significant; the

units abbreviations are wrong; eg, "11.39999999999991 sec" would better read "11.4 s". "ang" would better be replaced by "Å". Now about cosmetics: As it is right now, less than 20% of the space of the canvas contains useful information, while the rest is noise. The grey box hinders a lot of the visibility of the rhombohedral cell and the observation of diffusion. It would be better to show 1 unit cell, well-marked, and perhaps extend in other directions. Also to halve the size of y-coord and greatly reduce the z-coord. For instance, the panels shown in Figure 4 do a much better use of the space. All previous comments also apply to the GitHub repository.

Response 5

We appreciate this detailed comment. We found out that the axes in the videos and figures did not carry much meaning, and therefore we removed the axes and provided a zoomed-in version for all videos. We also greatly reduced the z-coordinate and rounded the time to three decimal places using the scientific notation. Please find the updated version on GitHub (https://github.com/VlachosGroup/Pdn-CO-Dynamics/tree/main/Simulation_Result_Graphics/Lattice_gifs) and in the submission package (Supplementary Movies 1-7).

Comment 6

* The author's notation is unnecessarily confusing. For instance, Pd₁, Pd₂, Pd₃ in Figure 2 (and elsewhere) are written without subscripts but yet referring to a monomer, a dimer, and a trimer, respectively. Why not use subscripts when such numbers refer to stoichiometry? Then, notations are sometimes not consistent, eg Pd₄-3D and Pd₄_3D, lines 201,205,207.

Response 6

Thank you for pointing this out. We used subscripts to refer to Pd_n clusters throughout the manuscript and SI and changed Pd₄-3D to Pd₄_3d. The changes can be found in:

Figure 2. Illustration of disparity in time scales of microscopic events.

Supplementary Figure 6. (a) Event frequency and (b) average time per event in the simulation with an initial state of 8 single atoms at 700 K, CO partial pressure of 0.1 bar at long times.

Supplementary Table 2. List of the elementary events included in the bare Pd_n system.

Index	Elementary event	Barrier E_a (e)	Prefactor A_{fwd} (s^{-1})	A_{ratio}
1	Pd ₁ diffusion on layer 0	0.14	1.00E+13	1
2	Pd ₁ diffusion on layer 1	0.05	1.00E+13	1
3	Pd ₁ diffusion on layer 2	0.05	1.00E+13	1
4	Pd ₁ diffusion on layer 3	0.05	1.00E+13	1
5	Pd ₂ diffusion on layer 0	0.37	1.00E+13	1
6	Pd ₃ diffusion on layer 0	0.71	1.00E+13	1
7	Pd ₄ diffusion on layer 0	1.35	1.00E+13	1
8	Pd _{4_3d} diffusion on layer 0	1.35	1.00E+13	1
9	Pd ₁ hopping from layer 0 to layer 1	0.85	1.00E+13	1
10	Pd ₁ hopping from layer 1 to layer 2	0.85	1.00E+13	1
11	Pd ₁ hopping from layer 2 to layer 3	0.85	1.00E+13	1

Supplementary Table 3. List of additional elementary events included in the Pd-CO system.

Index	Elementary event	Barrier E_a (e)	Pre-factor A_{fwd} (s^{-1})	A_{ratio}
12-20	CO adsorption on various sites	0	9.67E-02	5.20E-14
21	Pd ₁ (CO) diffusion on layer 0	0.62	1.00E+13	1
22	Pd ₂ (CO) diffusion on layer 0	0.52	1.00E+13	1
23	Pd ₃ (CO) diffusion on layer 0	1.17	1.00E+13	1
24	Pd _{4_3d} (CO) diffusion on layer 0	1.03	1.00E+13	1

Supplementary Figure 12. Elementary events in the bare Pd_n system. The top row indicates corresponding forward barrier in eV.

Supplementary Figure 17. Additional elementary events in the Pd-CO system. The top row indicates the corresponding forward barrier in eV.

Supplementary Figure 18. Average time per event (a) before and (b) after pre-factor scaling in the simulation with an initial state of 20 single atoms at 300K, CO partial pressure of 0.1 bar at long times (over 168 hours).

Supplementary Figure 19. Lattice snapshots and event frequency diagrams for each iteration in the simulation with an initial state of five single atoms at 300 K.

Comment 7

* Figure 2 and most Supp. Figures are missing units! Is the pressure in bar or mbar? Does "xi*" have units? It is not evident at a first sight what the caption "time evolution" is referring to without an intensive revision of all methods. How long was the sampling time for the "event frequencies"?

Response 7

Thank you for pointing this out. We use bar as the default unit for pressure throughout this work. We double-checked all figure to ensure that the units are included in the legends, axes, or captions. The following figures are updated:

Supplementary Figure 3. L1/L0 ratios for the systems starting with (a) 20 single atoms (b) 6 Pd₄_3d clusters.

Supplementary Figure 4. Final CO loadings (\bar{m}_{CO}) vs. L1/L0 ratio for systems starting with (a) single-atoms and (b) Pd₄_3d clusters.

"xi*" refers to the ratio between the number of adsorbed species and the number of Pd atoms, and therefore it has no unit. We replaced "time evolution" with "species ratios as a function of time" throughout the paper and SI. We rephrased the caption of figure 2 as below:

Figure 3. Species ratios as a function of time at 300 K exposed to a CO partial pressure $P_{CO}=0.1$ bar. (a) L1/L0 ratio - the number of Pd atoms in L1 layer to the number in the base L0 layer, \bar{x}_i^* - the ratio of total CO adsorbate counts on top, bridge, and hollow sites to the number of Pd atoms vs. time. (b) \bar{x}_i^* - the ratio of Pd atom counts on various Pd sites and (c)-(e) CO adsorption adsorbate counts on various CO adsorption sites to the number of Pd atoms vs. time.

The event frequencies in Figure 2 were sampled from simulations occurred over 168 clock hours. We revised the caption of Figure 2:

Figure 4. Illustration of disparity in time scales of microscopic events. (a) Scaled event frequency and (b) actual average time per event (without prefactor scaling) in a KMC simulation starting from a 0.05 coverage of single atoms at 300 K and $P_{CO}=0.1$ bar at long times (over 168 hours).

Comment 8

* Line 112.

* The authors jump too fast to the results and conclusions without first explaining how the pieces couple to each other. Only a mention to the Methods in line 90. Then the Methods section (L256-284) explains more what the authors did but not much about how the pieces fit together. They just refer the reader to Supplementary Figure 1 in line 256. Supplementary Figure 1 is actually the most important in all this work whose novelty is, precisely, the methods being used in this particular sequence. At least one paragraph detailing how the pieces fit together is needed when introducing the Results section. This is, *not* what the authors did, which is already well explained, but rather *why* they did it in that order, in few words.

Response 8:

Thank you for nice suggestion. We expanded the first paragraph of the Method section and moved it to the Results section. We showed the framework in Figure 1a-c and highlighted the connections between sub-modules and the motivations behind each decision (Page 3-4):

The multiscale framework (Figure 1a-c) allows modeling of metal clusters that are bare or exposed to an adsorbate used for characterization, e.g., CO in IR, or an intermediate created by the surface chemistry and does so at unprecedented computational efficiency. The framework integrates a comprehensive toolset including DFT calculations, genetic algorithm-based structure optimization, ML, and KMC. Details of all sub-models can be found in Supplementary Methods and prior works.^{10,23} We first investigate the electronic and structural characteristics of Pd_n clusters on CeO₂(111) and the CO adsorption energies on various sites using DFT. DFT calculations supply accurate data on metal cluster structures and energies. To overcome the computational cost of DFT and replace it, we train the ML-based Hamiltonian energy models (Figure 1a) on this DFT data as they represent an efficient structure-to-energy mapping. We develop two ML models using cluster expansion^{24,25} for pristine Pd clusters and the CO adlayer and represent the 3D structures (Figure 1d) on lattices. The lattices in Figure 1e, f consisting of well-defined sites for metal atoms (Pd) and adsorbates (CO), respectively. Combined with the Hamiltonian, Monte-Carlo-based structure optimization algorithms (Figure 1b), such a cluster genetic algorithm (CGA), predict the low energy structures at finite temperature. Active learning improves the model accuracy by passing the low energy structures for new DFT calculations and retraining the model iteratively. Next, we enumerate the possible elementary events, including diffusion of Pd single atoms and small clusters and CO adsorption/desorption, and compute their barriers in DFT. The Hamiltonian models and barriers parameterize the on-lattice KMC (Figure 1c) and enable thousands of on-the-fly calculations. Finally, the KMC engine executes key events and compute their transition rates to simulate the evolution of structures as a function of time.

Comment 9

Other minor points:

* The observation "single atoms do not disappear after agglomeration but always co-exist with clusters" is not fully new. It has also been observed for Pd on In₂O₃ (Nat Commun, 2019, 10, 3377). There, intermediate-size clusters are also active for CO₂ reduction to CO, whereas low-nuclearity ones are more selective to methanol.

Response 9:

Thank you for pointing us towards the right reference. We have added the following sentence to the discussion (Page 10):

The co-existence of single atoms and small clusters has also been observed for Pd_n on In₂O₃.⁴¹

The following reference was also added:

Frei, M. S. *et al.* Atomic-scale engineering of indium oxide promotion by palladium for methanol production via CO₂ hydrogenation. *Nat. Commun.* **10**, 1–11 (2019).

Comment 10

* The conclusion "low-temperature characterization data provides little insights into high-temperature catalysis" is perhaps an overstatement. A catalyst exposed to high temperatures, once major sintering/reconstruction has occurred, can then be characterized at room temperatures without large deviations. This is particularly true for single-atom and "extended" catalysts. The authors would better rephrase or, alternatively, show that the catalysts do change their behavior after annealing from high to room temperature.

Response 10:

Thank you for raising this excellent point. This was an overstatement. We rephrased the text to (Page 11):

low-temperature characterization data may be less relevant to the real-time high-temperature dynamics.